# NurViD: A Large Expert-Level Video Database for Nursing Procedure Activity Understanding

**Ming Hu**[1,2,3*]    **Lin Wang**[1,2,4*]    **Siyuan Yan**[1,3*]    **Don Ma**[1*]
**Qingli Ren**[5]    **Peng Xia**[1,2,3]    **Wei Feng**[1,2,3]
**Peibo Duan**[3]    **Lie Ju**[1,2,3]    **Zongyuan Ge**[1,2,3]

[1]AIM Lab, Faculty of IT, Monash University, Australia
[2]Airdoc-Monash Research, Airdoc, China
[3]Faculty of Engineering, Monash University, Australia
[4]College of Intelligent Systems Science and Engineering, Harbin Engineering University, China
[5]Nursing College, Shanxi Medical University, China

## Abstract

The application of deep learning to nursing procedure activity understanding has the potential to greatly enhance the quality and safety of nurse-patient interactions. By utilizing the technique, we can facilitate training and education, improve quality control, and enable operational compliance monitoring. However, the development of automatic recognition systems in this field is currently hindered by the scarcity of appropriately labeled datasets. The existing video datasets pose several limitations: 1) these datasets are small-scale in size to support comprehensive investigations of nursing activity; 2) they primarily focus on single procedures, lacking expert-level annotations for various nursing procedures and action steps; and 3) they lack temporally localized annotations, which prevents the effective localization of targeted actions within longer video sequences. To mitigate these limitations, we propose NurViD, a large video dataset with expert-level annotation for nursing procedure activity understanding. NurViD consists of over 1.5k videos totaling 144 hours, making it approximately four times longer than the existing largest nursing activity datasets. Notably, it encompasses 51 distinct nursing procedures and 177 action steps, providing a much more comprehensive coverage compared to existing datasets that primarily focus on limited procedures. To evaluate the efficacy of current deep learning methods on nursing activity understanding, we establish three benchmarks on NurViD: procedure recognition on untrimmed videos, procedure and action recognition on trimmed videos, and action detection. Our benchmark and code will be available at *https://github.com/minghu0830/NurViD-benchmark*.

## 1 Introduction

The application of deep learning (DL) in understanding nursing procedure activities has the potential to greatly enhance the quality of nurse-patient interactions, while also playing a crucial role in preventing medical disputes, minimizing missed nursing procedures, and reducing nursing errors [32, 45, 35, 37, 21, 11]. DL-based automatic approaches offer several key benefits for nurses, including: (1) *Reliable*: it enables objective, precise, and consistent assessments of nursing skills, eliminating the subjectivity inherent in traditional evaluations by experts [27]. (2) *Real-time guidance*: it facilitates immediate feedback on nurses' performance, empowering them to identify areas for improvement in a timely manner [30]. (3) *Cost-effective*: it can alleviate the burden of manual observations, evaluations, and training by experts [39].

---

[*]Equal Contribution

37th Conference on Neural Information Processing Systems (NeurIPS 2023) Track on Datasets and Benchmarks.

However, the development of automatic nursing recognition systems is currently hindered by the absence of suitably labeled datasets for nursing activities. Existing public video datasets for action recognition primarily focus on generic daily activities or specific sports, with minimal attention given to nursing or healthcare scenarios [23, 13, 36, 16, 22, 24, 17]. For example, the Kinetics 700 dataset [23] with 700 category labels includes only two labels related to nursing activities. Although some initiatives have attempted to create nursing activity understanding video datasets (e.g., handwashing-specific datasets [15, 7, 48, 42, 4, 26]), limitations arise due to the gap between them and real-world clinical settings, which is summarized as follows: (1) *Limited procedure and action variety:* Expensive annotation cost causes existing datasets only have a single nursing procedure, whereas real clinical environments involve a wide range of complex procedures. (2) *Simple scenes only:* The recorded videos are typically captured in controlled settings such as instructional or laboratory environments, which do not accurately reflect the complexity and variability of nursing procedures in actual clinical practice. (3) *Un-professional labeling:* They suffer from non-professional labeling or a lack of adherence to standard guidelines, leading to errors or inconsistencies. (4) *Short video sequence:* They primarily consist of short action clips, which do not facilitate understanding long-term activities and their context.

To mitigate these limitations, we proposed NurViD, a large-scale video benchmark for nursing procedure activity understanding. Compared to existing datasets, NurViD incorporates characteristics from the following aspects: (1) *Diverse procedure and action:* NurViD comprises 144 hours of annotated videos, which is approximately four times longer than the largest existing nursing activity datasets. It also contains 1,538 videos depicting 51 nursing procedure categories, covering the majority of common procedures, along with 177 action steps, providing much more comprehensive coverage, compared to previous datasets that primarily focus on single procedures with limited action steps. (2) *Real-world clinic settings:* Videos in NurViD were captured from over ten real clinical environments according to our statistics, including hospitals, clinics, and nursing homes. This diverse range of settings ensures that the models trained on NurViD are applicable in real-world clinical scenarios. (3) *Expert-level annotations:* NurViD was labeled by professionals with high expertise and knowledge in nursing. The procedure and action annotation process follows the guideline of *Training Outline for Newly Employed Nurses* issued by the *National Health Commission of China* [2], ensuring consistency and accuracy of the annotations. (4) *Support multiple recognition and detection tasks:* We have established two different classification tasks and an action temporal localization benchmark specifically targeting the long-tail distribution of the dataset.

We further compare our NurViD dataset with the other existing nursing activity video datasets and summarize the key difference in Table 1. The contributions of NurViD are summarized as follows:

- NurViD is the most diverse video benchmark to date for nursing procedure activity understanding tasks. It has been meticulously annotated by nursing professionals, and the annotation process follows standardized nursing procedure guidelines or protocols. NurViD exhibits a competitive video size and much more comprehensive coverage of procedures and action categories compared to existing datasets.

- In response to the practical needs of nursing and machine-learning communities, such as education and training, automatic action detection, and long-tail distribution, we establish three different recognition and localization benchmarks on NurViD.

- Long-term retention and availability of dataset. NurViD has been sourced from YouTube and follows the CC BY 4.0 license agreement [1].

## 2 Related Work

Research aimed at enabling machines to understand human behavior and activities has led to advancements in various practical applications. However, building such systems comes with challenges that require appropriate datasets for training and evaluation. In recent years, numerous datasets have been created to support research in human behavior understanding [34, 13, 49, 36]. While these datasets have been valuable for general activity recognition, only a limited number cater to the specific needs of nursing professionals.

**Sequential action prediction.** Standardized datasets have been meticulously designed for the purpose of evaluating the performance of algorithms in understanding and accurately recognizing intricate

| Datasets | Dataset Properties | | | | | | | | Tasks | | |
|---|---|---|---|---|---|---|---|---|---|---|---|
| | Publicly Available? | Expert-Level Annotations? | Real-World Settings? | No. of Videos | No. of Segments | No. of Procedures | No. of Actions | Total Duration | Procedure Recognition | Action Recognition | Action Detection |
| Llorca et al. [15] | ✗ | ✓ | ✗ | 8 | - | 1 | 7 | 4.2min | ✗ | ✓ | ✗ |
| Ameling et al. [7] | ✗ | - | ✗ | 24 | - | 1 | 6 | - | ✗ | ✓ | ✗ |
| Zhong et al. [48] | ✗ | - | - | 200 | 1,400 | 1 | 7 | - | ✗ | ✓ | ✗ |
| Wang et al. [42] | ✗ | ✗ | ✓ | 280 | 2,760 | 1 | 8 | - | ✗ | ✓ | ✗ |
| Kaggle [4] | ✓ | ✗ | ✗ | 292 | 3,504 | 1 | 12 | 23.3h | ✗ | ✓ | ✗ |
| Lulla et al. [26] | ✓ | - | ✓ | 3,185 | 6,689 | 1 | 7 | 38.9h | ✗ | ✓ | ✗ |
| NurViD (Ours) | ✓ | ✓ | ✓ | 1,538 | 5,608 | 51 | 177 | 144.4h | ✓ | ✓ | ✓ |

Table 1: The comparison among existing nursing procedure activity video datasets. Compared to other datasets, NurViD annotates the procedures and actions by following expert-level standards, focuses on more comprehensive coverage of various nursing procedure categories, collects a large number of videos, totaling 144 hours, and also enables action detection tasks.

activities within real-world scenarios [22, 16]. These datasets are used in various fields, including computer vision, robotics, natural language processing, and surveillance systems. Focusing on specific sets of actions carried out in a well-defined order, sequential action understanding datasets differ from those that encompass a broader range of contexts [49]. In many fields, it is essential to adhere to a strict sequence of steps to ensure optimal results. For example, in the medical field, following a specific order of steps is crucial in procedures such as administering medication, where any deviation from the established sequence could result in serious consequences for the patient. Standardized datasets can help ensure that the actions performed in real-world situations are accurately represented and provide a benchmark for evaluating the performance of algorithms.

**Nursing procedure video dataset.** Online learning has gained significant popularity, particularly through the utilization of instructional videos that provide step-by-step guidance, serving as valuable resources for teaching and learning specific tasks. Within the medical field, instructional videos have proven to be highly effective in conveying essential information using visual and verbal communication, thus benefiting learners [17]. On the other side, the absence of comprehensive and standardized nursing procedure video datasets presents a challenge in developing effective algorithms within the healthcare industry. The current datasets suffer from limited coverage, focusing only on nursing procedures relevant to specific healthcare settings or patient populations. Additionally, the quality of annotation plays a critical role in algorithm accuracy. The process of annotation involves identifying and labeling specific actions and events in the videos, which can be a time-consuming and challenging task. Annotation errors may arise due to factors such as human error, task ambiguity, or the absence of standardized protocols.

## 3 Building NurViD Dataset

In this section, we describe the process of building NurViD, from selecting the nursing procedures to acquiring, filtering, and annotating the video data. We leveraged the extensive collection of medical instructional videos available on YouTube [6] and carefully selected and filtered our video collection to ensure the quality and relevance of the dataset. We also developed a standardized labeling scheme for the actions performed in each video, providing a valuable resource for developing and evaluating algorithms that recognize and understand nursing procedures.

### 3.1 Procedure and Action Definition

The selection of nursing procedures and corresponding action steps is crucial for maintaining the relevance and usefulness of NurViD within the nursing profession and the broader healthcare community. The procedure selection adheres to a widely accepted nursing taxonomy, taking into account frequency of use and expert guidance. Then the actions involved in these procedures are gathered and standardized.

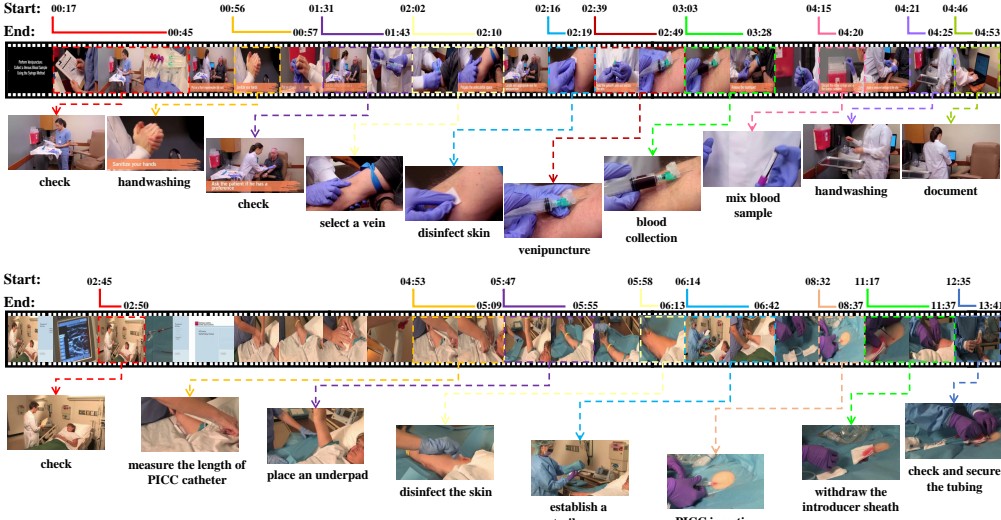

Figure 1: The examples for the annotated target action boundaries for *Intravenous Blood Sampling* and *Modified Seldinger Technique with Ultrasound for PICC Placement* procedures. The frames marked in colored boxes denote the annotated temporal boundaries for the target action steps.

**Procedure selection.** We compiled various nursing procedures from Nurselabs website [3] and nursing procedure books [8, 29, 31, 20]. With expert consultation, we identified 51 commonly employed nursing procedures that align with the requirements of most nursing scenarios. To validate their relevance and prevalence, we searched for corresponding videos on YouTube. Table 5 provides the full names and summarized abbreviations of these 51 procedures.

**Action definition.** We developed action steps for various nursing procedures based on college tutorials, and a nursing lecturer summarized the appropriate action labels by analyzing the action descriptions and video content. This was necessary for three main reasons: (1) There are variations in actions performed during specific nursing procedures, even within patients or instances of the same procedure, that require accurate representation of nuances. (2) The existence of diverse nursing procedure standards across countries and regions highlights the importance of establishing a unified standard. (3) Dealing with fine-grained video data from real-world nursing procedures requires rearranging action tags for the precise depiction of procedure nuances.

## 3.2 Online Video Crawling

Our objective in this stage was to gather sufficient videos demonstrating the pre-selected nursing procedures. By utilizing the extensive collection of medical instructional videos available on YouTube [6, 17], we acquire a wide range of videos without the need for third-party video production. To accomplish this, we queried YouTube using text-based searches for each procedure and obtained videos whose titles included the desired procedure keywords. To expand the video collection, we enhanced the search queries by including synonyms of each procedure. For example, *Subcutaneous Injection Insulin* can also be called *Subcutaneous Insulin Administration*, *Subcutaneous Insulin Therapy*, or abbreviated as *SCII*. Each video was downloaded at the highest resolution available. During video retrieval, we prioritized videos shorter than 20 minutes to limit the total storage.

## 3.3 Localization Annotation and Quality Control

In the NurViD dataset, each video is divided into multiple temporal segments, each of which contains only one action. Each action is annotated with its starting and ending timestamps as well as its frame position in the video. The annotation process is performed by undergraduates with medical and nursing backgrounds to ensure the accuracy and consistency of the annotations.

**Employing nursing professionals.** Data curation is an expensive process that typically involves extensive manual annotation. In some cases, datasets have employed a semi-automatic crowdsourcing approach for collection and annotation [12, 18, 40, 13]. For tasks that require greater reliability,

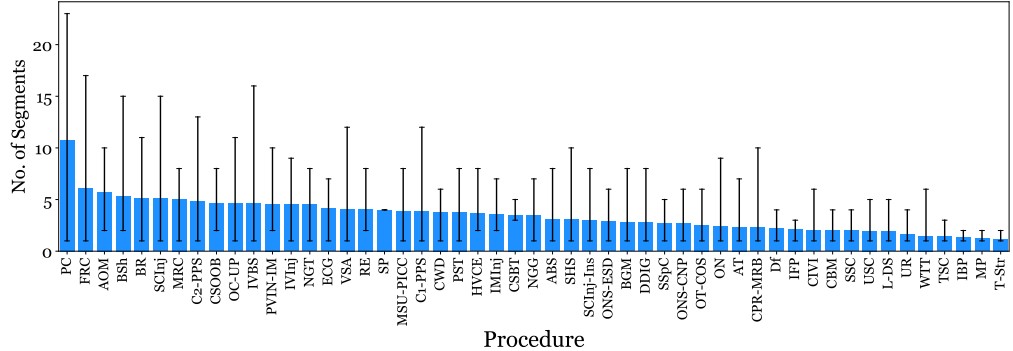

Figure 2: The average, maximum, and minimum number of action segments for each procedure.

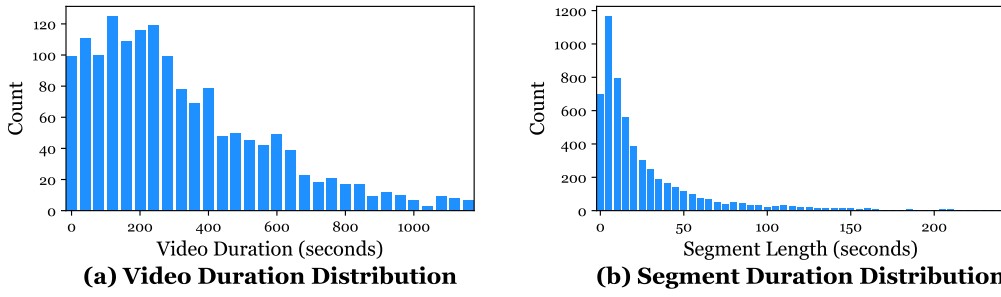

(a) Video Duration Distribution

(b) Segment Duration Distribution

Figure 3: NurViD dataset duration statistics.

certain datasets rely on domain experts for annotation, albeit at a higher cost. In our study, we formed a medical team of 26 individuals, consisting of a nursing lecturer and 25 nursing majors from a medical college, to perform labeling. Over half of the students have at least three years of undergraduate education, possess extensive practical experience in nursing procedures, and have successfully completed the university's standardized nursing procedure assessment.

**Invalid video filtering.** Additionally, certain videos may contain irrelevant content due to inaccuracies in text-based retrieval. Thus, in the first stage of labeling, we excluded videos that fall into the following categories: 1) showcasing unrealistic environments (e.g., movies, animation), 2) providing only verbal descriptions instead of visual demonstrations, 3) featuring static images instead of continuous videos, and 4) lacking the specified procedure.

**Action boundaries annotation.** To ensure localization annotation quality, we followed a three-round annotation process: (1) Each annotator was assigned 2-3 nursing procedures based on video count and tasked with filtering out inappropriate videos, (2) After filtering, the action segments of each procedure video were annotated by three members, (3) Finally, cross-checking of annotation results between every two groups was conducted to identify and rectify errors and omissions. This process resulted in a minimum of three annotated action boundaries for each video. To ensure reliable annotations, we employed the complete linkage algorithm [10] to cluster and merge various temporal boundaries into stable boundaries that received multiple agreements. It is important to mention that a single video may feature multiple separate instances of the target action, leading to multiple boundary definitions. Several examples of annotated target action boundaries are shown in Figure.1.

### 3.4 NurViD Statistics

Our NurViD dataset is a comprehensive collection of nursing procedure videos that includes 1,538 videos (144 hours) demonstrating 51 different nursing procedures and 177 actions for recognition and detection tasks, which are summarized in Table. 5 and Table. 6 in the supplementary material. To facilitate the development and evaluation of algorithms, we trimmed the videos based on annotated action boundaries, resulting in 5,608 trimmed video instances that totaled 50 hours. The trimmed videos have an average duration of 32 seconds, while the untrimmed videos have an average duration of 337 seconds. Over 74% videos have HD resolutions of 1280 × 720 pixels or higher. We observed a long-tailed distribution in the number of collected videos for both procedure and action.

# 4 Experimental Results

In our study, we focus on three tasks using the NurViD dataset: (1) procedure classification in untrimmed videos, (2) procedure and action classification on trimmed videos, and (3) action detection on untrimmed videos. To establish reliable baselines for these classification and detection tasks, we employ state-of-the-art models that have demonstrated effectiveness in human action recognition and detection. We provide a comprehensive analysis of the baseline models, taking into consideration the specific challenges posed by the NurViD dataset, such as long-tailed class distributions. For each task, we formulate the problem in detail and evaluate the performance of the baseline models. The results of our analysis can guide the development of more accurate and robust models for fine-grained action recognition and detection in healthcare applications.

## 4.1 Procedure Classification on Untrimmed Videos

This task involves identifying nursing procedures from untrimmed videos, which typically consist of multiple standardized action steps that must be executed in a specific order and other unrelated parts. The goal is to explore the effectiveness of DL technology in retrieving specific nursing procedures from a large video library. By accurately classifying nursing procedures, we can provide healthcare professionals with a powerful tool for quickly accessing relevant videos.

**Data settings.** The examples from each procedure category are randomly divided into three sets: 70% for training, 10% for validation, and 20% for testing, resulting in 1,054 training, 173 validation, and 311 testing videos, respectively.

**Class splits.** To account for the long-tailed nature [47, 43] of the NurViD dataset, we divided the procedure classes into three splits: *many*, *medium*, and *few*, based on the number of videos for each procedure. The accuracy of each split is the average accuracy of the included procedures within that split. Specifically, the *many* category includes the top 19.6% most frequent classes, the *medium* category includes the middle 43.1% classes, and the *few* category includes the remaining 37.3% classes. The number of classes per split is presented in Table 2.

| Baselines | Procedure Classification | | | |
| | Many | Medium | Few | All |
| | 10 | 22 | 18 | 50 |
|---|---|---|---|---|
| SlowFast [14] | 9.9 | 7.5 | 0.1 | 7.4 |
| C3D [38] | 10.7 | 5.1 | 1.8 | 7.7 |
| I3D [9] | 9.9 | 9.0 | 2.8 | 8.7 |
| SlowFast* | 19.9 | 10.2 | 5.0 | 13.5 |
| C3D* | **21.5** | 11.3 | **5.8** | **14.8** |
| I3D* | 19.8 | **12.5** | 5.6 | 13.1 |

Table 2: Per-class Top-1 accuracy for procedure prediction on untrimmed videos. The best performance for each split has been highlighted in **bold**.

**Baselines.** We compare the performance of SlowFast [14], I3D [9], and C3D [38] models on this task. These models are evaluated in two versions: 1) Random initialization training and 2) Pre-training with weights from Kinetics 400 [23], which is a human action recognition dataset.

**Results.** The results for per-class accuracy are summarized in Table 2. We find that C3D [38] is able to achieve competitive results for all the splits. However, the best top-1 per-class accuracy is 14.8%, indicating that there is significant room for improvement in this challenging task. We also observe that transfer learning from the model that is pre-trained on Kinetics 400 [23] improves the classification accuracy for all splits. With the C3D model, this corresponds to a per-class accuracy gain from 10.7% to 21.5% for the *many* category and from 5.1% to 11.3% for the *medium* category. Despite this improvement, accurately predicting procedure categories remains a significant challenge.

**Discussions.** Based on the results of the classification benchmarks established on NurViD, we found that even models pre-trained on Kinetics 400 [23] cannot achieve satisfactory classification performance. We speculate that this may be due to several reasons: (1) videos are not always exclusively focused on nursing procedure activities and may contain other unrelated content, such as verbal instructions and brief introductions to devices; (2) some actions, such as handwashing, disinfection, and document, are commonly used in various procedures, which may cause the model to obtain similar features in different procedural videos, making it difficult to classify them. Overall, focusing on the main procedure actions in the video remains a challenging task.

| Baselines | Procedure Classification | | | | Action Classification | | | | Joint Classification | | | |
|---|---|---|---|---|---|---|---|---|---|---|---|---|
| | Many | Medium | Few | All | Many | Medium | Few | All | Many | Medium | Few | All |
| | 13 | 21 | 17 | 51 | 9 | 66 | 87 | 162 | 17 | 78 | 224 | 302 |
| SlowFast [14] | 68.9 | 50.0 | 33.0 | 63.0 | 25.7 | 10.2 | 3.2 | 17.1 | 12.5 | 7.2 | 3.3 | 7.5 |
| C3D [38] | 70.1 | 48.8 | 33.0 | 63.9 | 22.9 | 9.3 | 2.9 | 15.9 | 13.8 | 7.3 | 3.5 | 7.7 |
| I3D [9] | 67.6 | 49.9 | 32.9 | 62.9 | 26.3 | 9.8 | 4.1 | 17.9 | 12.7 | 7.9 | 4.0 | 7.9 |
| SlowFast* | 71.2 | **61.8** | 39.0 | 68.9 | 29.8 | **15.5** | 7.9 | 21.1 | 21.2 | 9.4 | 5.6 | 12.8 |
| C3D* | **73.2** | 60.0 | 39.6 | **71.2** | 28.1 | 14.6 | 7.3 | **22.8** | **21.8** | **10.8** | **5.6** | **13.1** |
| I3D* | 70.7 | 60.4 | **40.9** | 70.0 | **31.3** | 14.8 | **8.2** | 21.5 | 19.5 | 9.9 | 4.7 | 12.5 |

Table 3: Per-class Top-1 accuracy (%) for the procedure, action, and their joint prediction on trimmed videos. * denotes the initialization from the model pre-trained on Kinetics 400 [23]. The best performance for each split has been highlighted in **bold**.

## 4.2 Procedure and Action Classification on Trimmed Videos

This task focuses on classifying both the primary action that occurs in a trimmed video and their associated procedures. By accurately classifying procedures and actions, building automated systems can automate the monitoring of each step in the nursing process, thereby helping to identify potentially missed diagnoses, nursing errors, and other issues. These systems can also help doctors and nurses quickly find useful nursing procedure videos, thereby improving their learning and work efficiency.

**Data settings.** The dataset was randomly partitioned, ensuring a balanced representation of examples from each procedure and action category. Specifically, we allocated 70% of the data for training (3,906 videos), 10% for validation (587 videos), and 20% for testing (1,122 videos).

**Class splits.** In this study, we also explore the long-tailed nature of the NurViD. We partitioned the videos into three subsets based on the task-specific distribution: *many*, *medium*, and *few*. For procedure categories, the breakdown is as follows: *many*: top 26% frequent classes, *medium*: middle 41% classes, and *few*: the remaining 33% classes. For action categories, *many*: top 5% frequent classes, *medium*: middle 37%, and *few*: the remaining 58% classes. In addition to some unique actions, such as *establish a sterile zone* that only exists in the *Modified Seldinger Technique with Ultrasound for PICC Placement* procedure, hand washing, skin disinfection, and other steps are common. Therefore, we established a joint classification task to explore the mutual influence between procedures and actions. For joint classification, *many*: top 5% frequent classes, *medium*: middle 21% classes, *few*: the remaining 74% classes. We show the number of classes per each split in Table 3.

**Baselines.** We compare the performance of three models, SlowFast [14], I3D [9], and C3D [38] on our tasks. These models are originally designed for human action recognition and lack the inherent ability to predict both a procedure and an action. To align with our joint prediction need, we introduce two task heads dedicated to procedure category recognition and action recognition, respectively. Consequently, we compute the joint loss, $\mathcal{L}_{joint}$, to handle these tasks. Hyper-parameters are tuned using the validation data, and the detailed hyper-parameter settings for each model can be found in the supplementary material. The loss is defined as $\mathcal{L}_{joint} = -\frac{1}{M}\sum_{i=1}^{M} y_i^p \log(p_i^p) - \frac{1}{N}\sum_{j=1}^{N} y_j^a \log(p_j^a)$ where $M$ is the number of procedure classes, $N$ is the number of action classes, $p_i^p$ and $y_i^p$ denote the procedure prediction probability and ground truth label for the category $i$, $p_j^a$ and $y_j^a$ denote the action prediction probability and ground truth label for the category $j$.

**Results.** The results are summarized in Table 3. All models perform well in the procedure classification task, with C3D[38] achieving a top-1 per-class accuracy of 71.2% across all splits. C3D[38] also demonstrates competitive performance in action and joint classification for all splits, with the best top-1 per-class accuracy of 22.8% and 13.1%, respectively. Transfer learning from a pre-trained model on Kinetics 400 [23] further improves the accuracy of procedure and action classification. For instance, using the C3D model, the per-class accuracy increases from 70.1% to 73.2% for *many* classification, from 48.8% to 60.0% for *medium* classification, and from 33.0% to 39.6% for *few* classification.

**Discussions.** The experimental results indicate that the performance of procedure classification on trimmed videos is significantly better than on untrimmed videos, which may confirm that irrelevant motion information has been filtered out in the trimmed videos, and the model is more likely to learn motion features. However, The imbalance in class frequencies poses difficulties in achieving

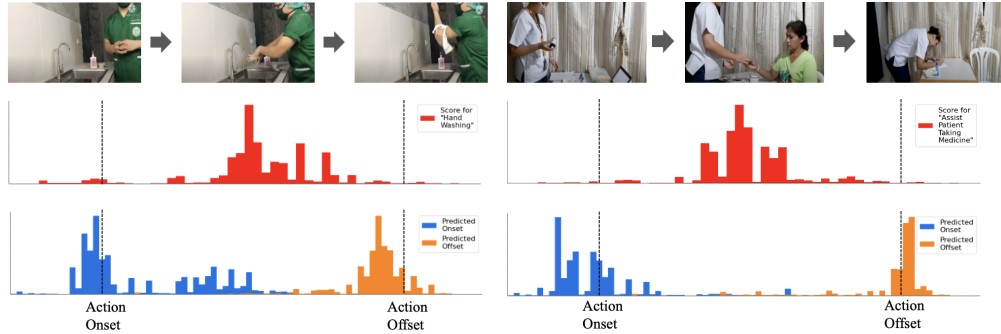

Figure 4: Visualization of action detection results. From top to bottom: (1) input video frames; (2) action scores at each time step; (3) histogram of action onsets and offsets computed by weighting the regression outputs using action scores.

satisfactory performance for those few classes. To alleviate it, further model design could incorporate long-tail learning techniques such as re-weighting or re-sampling techniques. These approaches can help mitigate the impact of class imbalance and improve the model's ability to generalize and classify the underrepresented classes more effectively.

## 4.3 Action Detection on Untrimmed Videos

The objective of this task is to accurately identify actions in untrimmed videos by determining the temporal extent of the main activity. To establish a benchmark for this task, we adopt three baseline models [46, 28, 33] that have shown effectiveness in temporal action localization [13, 46]. The evaluation metric used is the mean Average Precision (mAP), calculated at various temporal Intersection over Union (tIoU) thresholds [0.5:0.1:0.9]. Additionally, we provide the average mAP across different tIoUs.

**Data settings.** Following previous approaches for the standard procedure and action classification, we adhere to a split ratio of [train: 0.7, val: 0.1, test: 0.2] to divide the untrimmed videos at the procedure-action composition level. Consequently, we have 1,077 videos for training, 153 for validation, and 308 for testing.

**Baselines.** To evaluate the performance of our datasets, we utilize baseline models, namely Action-Former [46], TAGS [28], and TriDet [33]. In order to generate features for the NurViD videos, we fine-tune a two-stream I3D model (I3D [9]) initially pre-trained on ImageNet [12]. Subsequently, we extract RGB and optical flow features for each video and concatenate them as the model input.

**Results.** We show the action detection results in Table 4. Among the baselines, ActionFormer [46] achieves the highest performance, with an average mAP of 23.9% and an mAP of 32.9% for the threshold of 0.5.

**Discussions.** The outputs of the Action-Former [46] model are visualized in Figure. 4. These outputs consist of action scores and regression results, which are weighted by the action scores and presented as a weighted histogram. Nursing actions, unlike actions in natural datasets,

| | mAP (%) | | | | | |
|---|---|---|---|---|---|---|
| Baselines | 0.5 | 0.6 | 0.7 | 0.8 | 0.9 | Avg. |
| TriDet [33] | 30.3 | 26.7 | 24.3 | 20.1 | 10.7 | 20.8 |
| TAGS [28] | 31.4 | 26.5 | 22.6 | 19.2 | 11.5 | 22.4 |
| ActionFormer [46] | **32.9** | **29.6** | **25.8** | **20.8** | **12.7** | **23.9** |

Table 4: The results of action detection. We report mAP at the IoU thresholds of [0.5:0.1:0.9]. The average mAP is calculated by averaging the mAP scores across various tIoU thresholds.

involve precise and meticulous movements rather than large-scale motion from frame to frame. Therefore, a more detailed action representation is necessary to accurately describe and learn nursing actions. For instance, in the hand wash procedure recommended by the WHO [5], it is crucial to consider small changes in hand position and environmental factors. Since these movements are extremely subtle, algorithms capable of handling finer granularity can potentially detect these subtle motion changes.

# 5    Limitations

We believe that NurViD is beneficial for advancing the development of AI technology in the nursing field. However, we must also consider the potential risks and impacts that may arise from anticipated or foreseeable applications. Additionally, NurViD is downloaded from diverse YouTube channels, video quality, production style, and regional nursing practice differences can introduce biases in nursing procedure representation.

**Intended/Foreseeable Uses.** After reviewing relevant literature and discussing with nursing professors and students, we have noticed two main challenges in nursing training and learning: (1) the issue of imprecise recommendations encountered by students when searching for learning videos. For example, when we want to learn about Intravenous Injection procedure videos on YouTube, due to the limitations of the recommendation algorithm, the website may recommend Intravenous Blood Sampling procedure videos to some extent, which is quite common. However, the system trained on our dataset can provide more accurate classification recommendations for searches related to nursing operations. Additionally, since NurViD includes temporal localization annotations for actions, it means that we can dynamically adjust the playback range based on our specific interests, for instance, if I only need to watch the step of skin disinfection and not the other steps, I just need to enter the text "skin disinfection," and the system will automatically locate the specific segment for me. This undoubtedly saves some unnecessary time wastage, and (2) in China alone, there are over 200,000 undergraduate nursing students each year, and this number continues to grow. Each student is required to pass a professional nursing skills examination and undergo approximately 1000 to 2000 hours of practical training. However, currently, students still heavily rely on experienced teachers for real-time supervision and feedback during training, which requires a significant amount of human resources and time investment. (3) Real-time monitoring: NurViD is a dataset that leans more towards general models and is currently primarily used for testing and improving deep learning models. It aims to assist students' learning and training by recording, monitoring, and providing feedback during their practice sessions, reducing the need for teaching resources, and facilitating event documentation. In this mode, even rough feedback can save a significant amount of costs. However, AI systems cannot guarantee absolute accuracy, which means that detection errors or omissions may occur. Therefore, it is strongly recommended that any system built upon NurViD or similar technologies clearly communicate their limitations and potential errors from the outset and appropriately incorporate human assistance during usage.

**Potential Privacy.** In human-action understanding of video datasets, it is often inevitable to encounter faces. Therefore, we provide a script that uses OpenCV's Haar classifier to detect the facial regions in videos and blur them. We will implement more advanced methods to further alleviate the privacy problem in the future.

**Employment Risks.** The employment risks associated with nursing action recognition systems can involve the following aspects: **(1) Reduced Workforce Demand:** The integration of nursing action recognition systems may lead to a diminished need for human nursing professionals. The incorporation of automation technology can handle routine nursing tasks, thereby potentially decreasing the reliance on human caregivers. Consequently, this could result in certain nursing professionals facing job scarcity or encountering uncertainty in their employment prospects. **(2) Shift in Skill Requirements:** Nonetheless, our perspective is that the future implementation of nursing action recognition systems will likely require nursing professionals to acquire new skills and knowledge to adeptly interact with the technology rather than being replaced by it. This could involve gaining proficiency in various areas, such as effectively engaging with the system, accurately interpreting its outputs, and promptly addressing any discrepancies that arise. Nursing professionals encountering challenges in adapting to these evolving technological demands might find it necessary to undergo retraining efforts. **(3) Technical malfunctions and misidentification risks:** Nursing action recognition systems come with inherent risks of technical glitches and misidentifications. Inaccuracies or erroneous assessments made by the system could lead to misguided nursing judgments or actions. This potentially jeopardizes patient safety and well-being, obliging nursing professionals to dedicate extra time and effort toward rectifying system errors. To mitigate the employment risks linked to nursing action recognition systems, a comprehensive approach is crucial: (1) Workforce Enhancement Programs: Offering programs for upskilling and transitioning is vital to empower nursing professionals with the competencies needed to navigate evolving roles in tandem with the technology. (2) Ethical Guidelines and Standards: Establishing clear ethical guidelines ensures

responsible and morally sound utilization of nursing action recognition systems within healthcare settings. (3) Data Privacy and Security: Implementing robust measures to safeguard patient data and privacy is paramount to engender trust in the system's operation. (4) Human Oversight and Decision-making: Maintaining human oversight ensures that critical nursing decisions are grounded in human judgment and understanding, acting as a safeguard against erroneous system outputs. (5) Continuous System Evaluation and Enhancement: Regularly evaluating and enhancing the system's performance is pivotal to addressing any technical shortcomings and refining its accuracy. (6) Stakeholder Engagement and Collaboration: Fostering engagement and collaboration among various stakeholders, including nursing professionals, technologists, and policymakers, promotes a holistic approach to system development and implementation.

**Contestability/Explainability Issues.** It is important to acknowledge that while AI systems can assist in detecting standardized nursing procedures and actions, they are not infallible and cannot achieve perfect accuracy. Human supervision and oversight are indispensable in ensuring patient safety and quality care. Therefore, it is highly recommended that any system developed based on NurViD or similar technologies explicitly state their limitations and potential errors upfront. This can be achieved by providing clear disclaimers, agreements, or warnings to users, emphasizing the need for human involvement, critical thinking, and professional judgment when interpreting and acting upon system outputs. By transparently communicating the system's limitations, healthcare professionals can make informed decisions and use the technology as a supportive tool rather than relying solely on its outputs.

**Potential Regional Biases.** Standardization poses a significant challenge due to the diverse origins of the videos and the variation in nursing procedure guidelines across countries. NurViD aims to cover almost all common action labels, providing flexibility for different regions to adopt their own standards based on it. However, despite our efforts, it is important to acknowledge that complete avoidance of bias is challenging. Additionally, the comprehensiveness of the dataset may be influenced by the sources and origins of the videos. The video collection process for NurViD may inadvertently introduce biases towards certain regions or healthcare settings. This bias can limit the generalizability of the dataset to a broader context, as it may not fully capture the diverse range of nursing procedures and actions practiced worldwide. Addressing this limitation requires ongoing efforts to collect data from diverse regions, collaborate with experts from different backgrounds, and ensure a balanced representation of nursing practices from various healthcare contexts. Furthermore, domain adaptation models may also be one of the potential solutions [41, 44, 43].

**Comprehensiveness of Nursing Procedures and Actions.** While efforts were made to include a wide range of common action labels, it is important to acknowledge that the dataset may not cover every possible nursing procedure or action. Variations in nursing practices and guidelines across different regions and healthcare systems can result in some actions being omitted or not adequately represented in the dataset. Furthermore, the dataset's composition may be influenced by the availability and accessibility of videos from different regions. Certain nursing procedures or actions that are more prevalent or emphasized in specific regions may be overrepresented, while others may be underrepresented. Continuously expanding the dataset's coverage through collaboration with experts and professionals from diverse backgrounds can help address this limitation and enhance its comprehensiveness.

## 6   Conclusion

We introduce NurViD, a comprehensive video dataset designed for nursing procedure activity understanding. By collecting videos from YouTube and meticulously annotating action sequences at an expert-level, NurViD offers a rich resource for studying nursing procedures. The dataset encompasses 1,538 untrimmed videos, each averaging 32 seconds in duration, covering 51 distinct procedures and 177 action steps. We present three tasks based on NurViD: procedure classification on untrimmed videos, procedure and action classification on trimmed videos, and action detection on untrimmed videos. Our experiments demonstrate that accurate recognition of procedures and the action steps they contain is challenging even with current state-of-the-art models, particularly when the dataset exhibits a long-tail distribution. To promote further development in the field of nursing procedure analysis, we will release all of our data and code to the public, enabling researchers to build upon our work and advance the understanding of nursing activities.

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

# A  NurViD Statistics Supplement

The distribution of data for each procedure and action is depicted in Figure.5 and Figure.6, respectively. Our findings reveal a long-tail distribution for the number of collected and trimmed videos per type.

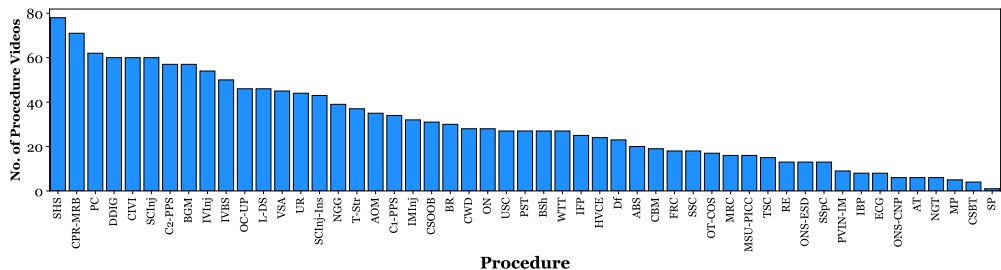

Figure 5: Number of untrimmed videos per each procedure category. We rank the categories according to their untrimmed video frequency.

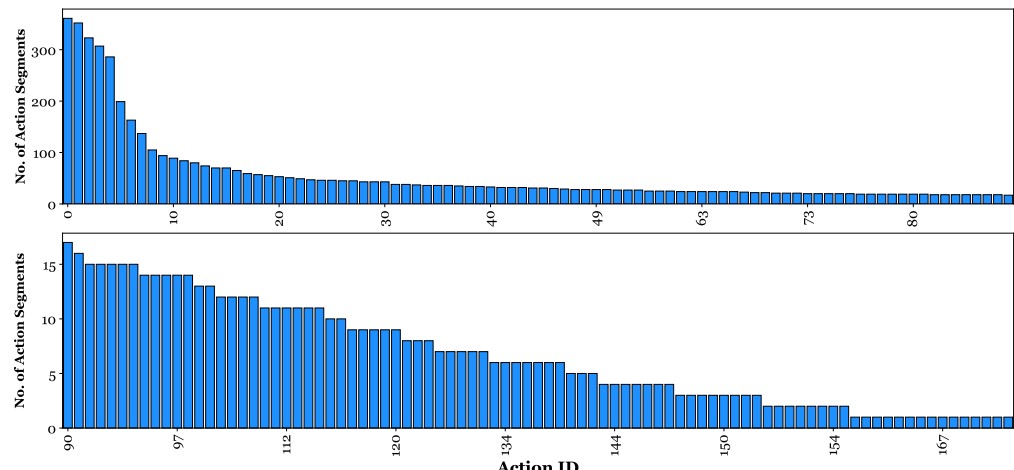

Figure 6: Number of trimmed videos per each action category. We rank the categories according to their trimmed video frequency.

# B  Training Details

We conducted the classification experiments using four NVIDIA RTX3090Ti GPUs and the detection experiments using four NVIDIA RTX A6000 GPUs. All of our code can be found attached in the supplementary material and also on the project homepage *https://github.com/minghu0830/NurViD-benchmark*.

## B.1  Training Details for Procedure and Action Classification

We employed the officially released codes to train all recognition models. The SlowFast [14] and I3D [9] models were trained for 196 epochs with a batch size of 64, using a base learning rate of 0.1. We employed a cosine decay learning rate scheduler with 34 warm-up epochs. We sampled 16 frames per clip with a sampling rate of 24. For the C3D [38], we used a base learning rate of 0.1, a cosine decay learning rate scheduler, trained for 196 epochs, with 34 warm-up epochs and a batch size of 32. We sampled 16 frames per clip with a sampling rate of 24.

## B.2  Training Details for Action Detection

*Feature extraction.* To extract features from the videos, we first extracted RGB frames from each video at a rate of 25 frames per second. We also extracted optical flow using the TV-L1 [19, 25] algorithm. We then fine-tuned an I3D [9] model that had been pre-trained on the ImageNet [12] dataset, and used it to generate features for each RGB and optical flow frame. Because each video has a variable duration, we performed uniform interpolation to

generate 100 fixed-length features for each video. Finally, we concatenated the RGB and optical flow features into a 2048-dimensional embedding, which served as the input for our model.

*Model training.* We trained all detection models using their officially released code and default configurations. For training the ActionFormer [46] model, we used a base learning rate of 0.001, a cosine decay learning rate scheduler, trained for 30 epochs with 5 warmup epochs, and a batch size of 16. For the TAGS [28] model, we used a base learning rate of 0.0004, a step decay learning rate scheduler, trained for 20 epochs, and a batch size of 200. For the TriDet [33] model, we used a base learning rate of 0.0001, trained for 50 epochs, and a batch size of 256.

## C The Long-tail Distribution of Procedure, Action, and Their Composition

Our data has a skewed distribution in terms of the procedure, action, and also their composition. To get better insights, we split the categories into *many*, *medium*, and *few* groups based on their frequency. We show the procedure distribution in Figure. 7 and Figure. 8, the action distribution in Figure. 9 and their compositional distribution in Figure. 10.

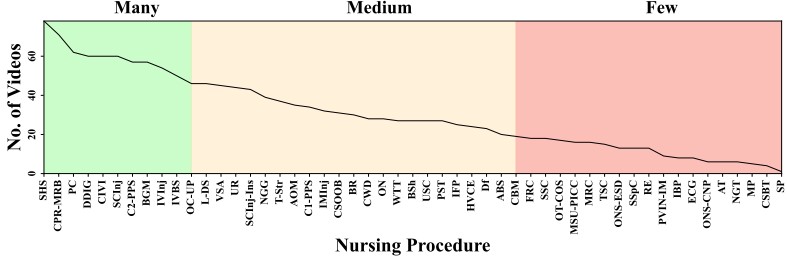

Figure 7: *The number of untrimmed videos per each procedure.* The procedures with the frequency ≥ 50 are grouped into *many*. The procedures with the frequency < 50 and ≥ 20 are grouped into *medium*. The procedures with the frequency < 20 are grouped into *few*. We rank the procedures based on their frequency.

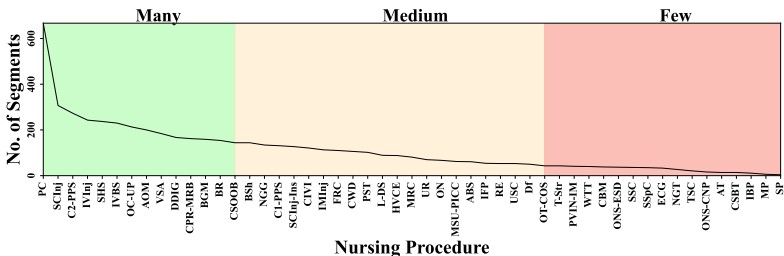

Figure 8: *The number of trimmed videos per each procedure.* The procedures with the frequency ≥ 150 are grouped into *many*. The procedures with the frequency < 150 and ≥ 45 are grouped into *medium*. The procedures with the frequency < 45 are grouped into *few*. We rank the procedures based on their frequency.

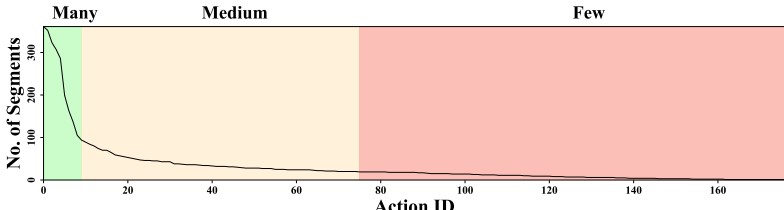

Figure 9: *The number of trimmed videos per each action.* The actions with the frequency ≥ 100 are grouped into *many*. The actions with the frequency < 100 and ≥ 20 are grouped into *medium*. The actions with the frequency < 20 are grouped into *few*. We rank the actions based on their frequency.

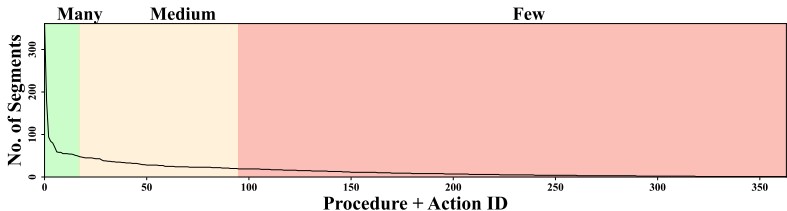

Figure 10: *The number of trimmed videos per each composition of procedure category and action.* The compositions with the frequency $\geq 50$ are grouped into *many*. The compositions with the frequency $< 50$ and $\geq 20$ are grouped into *medium*. The compositions with the frequency $< 20$ are grouped into *few*. We rank the compositions based on their frequency.

# D Compositional Low-shot Procedure and Action Classification on Trimmed Videos

Obtaining an adequate number of labeled action samples for all procedures in our collected taxonomy poses challenges. To address this, we plan to propose some compositional low-shot procedure and action classification tasks (0-shot, 1-shot, and 5-shot) to investigate these phenomena in the future.

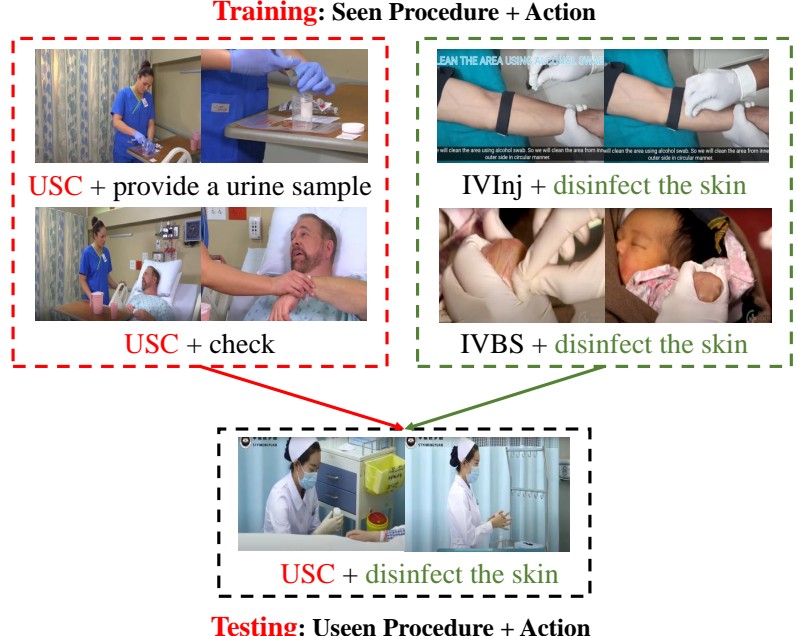

Figure 11: Demonstration of compositional zero-shot procedure and action recognition.

# E Annotation Interface Demonstration

The annotation work in the early stages is mainly divided into two phases: (1) procedure verification; (2) action localization annotation. In the first phase, annotators need to check the information prompted by the interface, judge whether the video belongs to the nursing procedure, and filter out unqualified videos such as animations, voice broadcasts, and slideshows through the skip button. In the second phase, the annotation mainly focuses on the action localization of the qualified videos screened in the first phase. The two frame windows correspond to the starting frame and ending frame of the annotated segment.

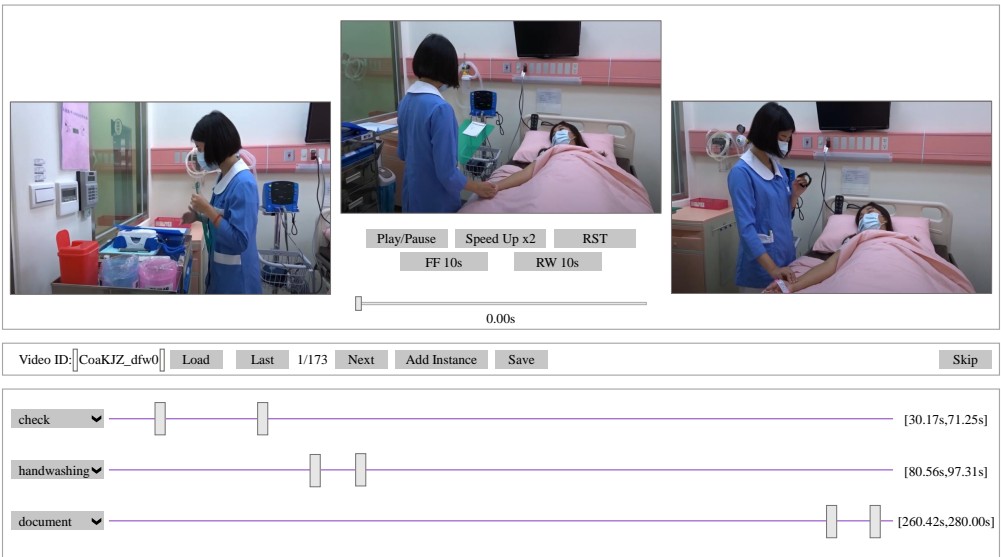

Figure 12: The procedure verification and action localization interface.

| ID | Procedure | Abbreviation |
|---|---|---|
| 0 | Surgical Hand Scrub | SHS |
| 1 | Cardiopulmonary Resuscitation WIth Manual Resuscitation Bag | CPR-MRB |
| 2 | Perineal Care | PC |
| 3 | Donning and Doffing Isolation Gowns | DDIG |
| 4 | Closed Intravenous infusion | CIVI |
| 5 | Subcutaneous Injection | SCInj |
| 6 | Change a Two-Piece Pouching System | C2-PPS |
| 7 | Blood Glucose Monitoring | BGM |
| 8 | Intravenous Injection | IVInj |
| 9 | Intravenous Blood Sampling | IVBS |
| 10 | Oral Care for Unconscious Patients | OC-UP |
| 11 | Logrolling with Draw Sheet | L-DS |
| 12 | Vital Sign Assessment | VSA |
| 13 | Use of Restraints | UR |
| 14 | Subcutaneous Injection Insulin | SCInj-Ins |
| 15 | Nasogastric Gavage | NGG |
| 16 | Transfer with Stretcher | T-Str |
| 17 | Administering Oral Medications | AOM |
| 18 | Change a One-Piece Pouching System | C1-PPS |
| 19 | Intramuscular Injection | IMInj |
| 20 | Change Sheets of an Occupied Bed | CSOOB |
| 21 | Bed Rubbing | BR |
| 22 | Change Wound Dressings | CWD |
| 23 | Oxygen Nebulization | ON |
| 24 | Wheelchair Transfer Technique | WTT |
| 25 | Bed Shampoo | BSh |
| 26 | Urine Specimen Collection | USC |
| 27 | Penicillin Skin Testing | PST |
| 28 | Infusion by Pump | IFP |
| 29 | High-Volume Colonic Enemas | HVCE |
| 30 | Defibrillation | Df |
| 31 | Arterial Blood Sampling | ABS |
| 32 | Closed Bed Making | CBM |
| 33 | Female Retention Catheterization | FRC |
| 34 | Stool Specimen Collection | SSC |
| 35 | Oxygen Therapy with Central Oxygen Supply | OT-COS |
| 36 | Male Retention Catheterization | MRC |
| 37 | Modified Seldinger Technique with Ultrasound for PICC Placement | MSU-PICC |
| 38 | Throat Swab Collection | TSC |
| 39 | Oral and Nasal Suctioning with Electric Suction Device | ONS-ESD |
| 40 | Sputum Specimen Collection | SSpC |
| 41 | Retention Enema | RE |
| 42 | Peripheral Venous Indwelled Needle Infusion and Maintaince | PVIN-IM |
| 43 | Injection by Pump | IBP |
| 44 | Electrocardiogram | ECG |
| 45 | Oral and Nasal Suctioning with Central Negative Pressure Device | ONS-CNP |
| 46 | Aseptic Technique | AT |
| 47 | Nasogastric Tube | NGT |
| 48 | Multi-Parameter Monitoring | MP |
| 49 | Closed System Blood Transfusion | CSBT |
| 50 | Skin Preparation | SP |

Table 5: The procedure names and their abbreviations in this paper.

| ID | Action | ID | Action |
|----|--------|----|--------|
| 0 | Clean and scrub the perineum | 89 | Measure respiration |
| 1 | Disinfect skin | 90 | Change upper clothing |
| 2 | Handwashing | 91 | Wear gloves |
| 3 | Position the patient | 92 | Collect pharyngeal swab specimen |
| 4 | Check | 93 | Two-person transfer |
| 5 | Inject medication | 94 | Adjust drip rate |
| 6 | Place an underpad | 95 | Cleanse inner surfaces of teeth |
| 7 | Venipuncture | 96 | Connect lead wires |
| 8 | Cleanse skin | 97 | Cleanse inner surfaces of teeth |
| 9 | Rinse with running water | 98 | Collect sputum specimen |
| 10 | Document | 99 | Prepare operating space |
| 11 | Secure ostomy bag | 100 | PICC insertion |
| 12 | Perform surgical hand scrub | 101 | Remove the base plate |
| 13 | Blood collection | 102 | Connect suction catheter |
| 14 | Prepare medication solution | 103 | Loosen isolation gown |
| 15 | Release trapped air | 104 | Perform oral-pharyngeal suction |
| 16 | Select a vein | 105 | Remove urinary catheter |
| 17 | Immobilize the shoulder | 106 | Install oxygen inhalation device |
| 18 | Remove needle | 107 | Assist with bed rest |
| 19 | Perform subcutaneous puncture | 108 | WWithdraw the introducer sheath |
| 20 | Measure blood glucose level | 109 | Check the thermometer |
| 21 | Remove ostomy bag | 110 | Perform surgical hand disinfection |
| 22 | Connect infusion device | 111 | Fasten buckle |
| 23 | Remove isolation gown | 112 | Shift to the right side |
| 24 | Apply leak prevention ointment | 113 | Organize the bed unit |
| 25 | Prepare glucometer | 114 | Perform three-person transfer |
| 26 | Perform chest compressions | 115 | Cover pillow with pillowcase |
| 27 | Assist with ventilation using a simple respirator | 116 | Select the needle puncture site |
| 28 | Draw bed curtains | 117 | Cleanse perineum |
| 29 | Turn patient to left lateral position | 118 | Comb hair |
| 30 | Trim ostomy bag baseplate | 119 | Inspect urinary catheter |
| 31 | Establish a sterile zone | 120 | Perform four-person transfer |
| 32 | Spread the proximal bedsheet | 121 | Adjust negative pressure |
| 33 | Rinse shampoo | 122 | Secure urinary catheter |
| 34 | Insert urinary catheter | 123 | Observe skin around wound site |
| 35 | Measure blood pressure | 124 | Observe results of skin test |
| 36 | Apply skin protection film | 125 | Change pillowcase |
| 37 | Put on isolation gown | 126 | Flush the sealed tube |
| 38 | Insert rectal tube | 127 | Withdraw nebulizer |
| 39 | Prepare medications | 128 | Secure the indwelling needle |
| 40 | Apply shampoo | 129 | Soak feet |
| 41 | Perform seven-step handwashing technique | 130 | Remove isolation gown |
| 42 | Aspirate medication | 131 | Dispose of arterial blood collection device |
| 43 | Assist patient taking medicine | 132 | Cleanse cheeks |
| 44 | Measure body temperature | 133 | Check the blood pressure meter |
| 45 | Set parameters | 134 | Replace clean bedsheet |
| 46 | Measure pulse | 135 | Move and transfer |
| 47 | Remove proximal bedsheet | 136 | Evaluate wound status |
| 48 | Transport in wheelchair | 137 | Rinse suction catheter |
| 49 | Fill in dressing | 138 | Expose the connection sit |
| 50 | Rub upper limbs | 139 | Evaluate resuscitation effect |
| 51 | Install nebulizer | 140 | Connect the monitor |
| 52 | Wash face | 141 | Measure the length of PICC catheter |
| 53 | Insert gastric tube | 142 | Perform nasopharyngeal and nasotracheal suction |
| 54 | Spread the opposite side bed sheet | 143 | Observe drainage situation |
| 55 | Identify cardiac arrest | 144 | Save electrocardiogram (ECG) results |
| 56 | Remove rectal tube | 145 | Remove the lead wires |
| 57 | Perform intradermal puncture | 146 | Perform single-person transfer |
| 58 | Moisten hair | 147 | Connect suction tube |
| 59 | Prepare defibrillation device | 148 | Prepare cotton balls |
| 60 | Dry hair | 149 | Cleanse hard palate |
| 61 | Confirm the position of the gastric tube in the stomach | 150 | Cleanse tongue surface |

| ID | Action | ID | Action |
|---|---|---|---|
| 62 | Secure the base | 151 | Connect injection device |
| 63 | Cleanse lips | 152 | Withdraw contaminated bed shee |
| 64 | Defibrillate | 153 | Transfuse blood |
| 65 | Cleanse chest and abdomen | 154 | Withdraw oxygen inhalation device |
| 66 | Cleanse back | 155 | Remove gastric tube |
| 67 | Open airway | 156 | Organize the blood pressure mete |
| 68 | Cleanse outer surfaces of teeth | 157 | Disinfect instruments |
| 69 | Adjust oxygen flow rate | 158 | Change dressing |
| 70 | Tie waist knot | 159 | Observe defibrillation results |
| 71 | Dry hands | 160 | Take treatment towels |
| 72 | Withdraw the opposite side bed sheet | 161 | Apply conductive gel |
| 73 | Measure the length of the gastric tube | 162 | Cover with bed sheet |
| 74 | Spray stoma care powder | 163 | Rinse mouth |
| 75 | Defibrillate | 164 | Change pants |
| 76 | Remove dressing | 165 | Secure drainage tube |
| 77 | Perform arterial puncture | 166 | Check medication |
| 78 | Collect urine specimen | 167 | Take treatment bowl |
| 79 | Spread the large sheet | 168 | Pour sterile solution |
| 80 | Administer oxygen | 169 | Cleanse oral cavity bottom |
| 81 | Guide nebulization | 170 | Restrict knee |
| 82 | Check and secure the tubing | 171 | Mark |
| 83 | Rub lower limbs | 172 | Monitor blood oxygen saturation |
| 84 | Mix blood sample | 173 | Check the pressure reducer |
| 85 | Prepare skin test solution | 174 | Inspect skin |
| 86 | Secure gastric tube | 175 | Perform endotracheal suctioning |
| 87 | Nasogastric feeding | 176 | Monitor electrocardiogram (ECG) |
| 88 | Collect stool specimen | | |

Table 6: 177 actions in this paper.

