**For what purpose was the dataset created?** Was there a specific task in mind? Was there a specific gap that needed to be filled? Please provide a description.

The application of deep learning to nursing procedure activity understanding has the potential to greatly enhance the quality and safety of nurse-patient interactions. By utilizing the technique, we can facilitate training and education, improve quality control, and enable operational compliance monitoring. The existing video datasets had several limitations, including small scale, focus on single procedures, and lack of temporally localized annotations. To fill this gap, our dataset was developed as a large video dataset with expert-level annotations for nursing procedure activity understanding.

**Who created this dataset (e.g., which team, research group) and on behalf of which entity (e.g., company, institution, organization)?**

The dataset was created by AIM for Health Lab at Monash University.

**Who funded the creation of the dataset?** If there is an associated grant, please provide the name of the grantor and the grant name and number.

No funding.

**Any other comments?** No other comments.

---

**Composition**

**What do the instances that comprise the dataset represent (e.g., documents, photos, people, countries)?** Are there multiple types of instances (e.g., movies, users, and ratings; people and interactions between them; nodes and edges)? Please provide a description.

The dataset is composed of videos from YouTube. The video content involves 51 nursing operations being performed in different environments such as hospitals, clinics, and other settings.

**How many instances are there in total (of each type, if appropriate)?**

There are 1,538 long videos in total, which can be divided into 5,615 video clips.

**Does the dataset contain all possible instances or is it a sample (not necessarily random) of instances from a larger set?** If the dataset is a sample, then what is the larger set? Is the sample representative of the larger set (e.g., geographic coverage)? If so, please describe how this representativeness was validated/verified. If it is not representative of the larger set, please describe why not (e.g., to cover a more diverse range of instances, because instances were withheld or unavailable).

The NurViD dataset is a sample of instances from a larger set of videos available on YouTube. The dataset may be somewhat representative of videos from these countries, but it is unclear how representative it is of videos from other countries or regions. The NurViD dataset was not designed to be a representative sample of all videos available on YouTube. Instead, it was created to provide a diverse and naturalistic sample of videos for use in research on video analysis and understanding.

**What data does each instance consist of? "Raw" data (e.g., unprocessed text or images) or features?** In either case, please provide a description.

Each instance consists of a corresponding annotation in the JSON file, which contains labels of procedures, actions, and their localization annotation.

**Is there a label or target associated with each instance?** If so, please provide a description.

Each instance is labeled with its procedure and action category, as well as the temporal localization interval of the action, including parameters such as the video's resolution, fps, total duration, etc.

**Is any information missing from individual instances?** If so, please provide a description, explaining why this information is missing (e.g., because it was unavailable). This does not include intentionally removed information, but might include, e.g., redacted text.

The annotation format and information for all instances are consistent and complete, with no missing data.

**Are relationships between individual instances made explicit (e.g., users' movie ratings, social network links)?** If so, please describe how these relationships are made explicit.

No.

**Are there recommended data splits (e.g., training, development/validation, testing)?** If so, please provide a description of these splits, explaining the rationale behind them.

For different tasks, we split long videos or action segments into training, validation, and testing sets according to different ratios.

**Are there any errors, sources of noise, or redundancies in the dataset?** If so, please provide a description.

It is inevitable that there will be errors or noise in any dataset, especially in manually annotated datasets. However, we take as many measures as possible to avoid this phenomenon. To ensure localization annotation quality, we followed a three-round annotation process: (1) Each annotator was assigned 2-3 nursing procedures based on video count and tasked with filtering out inappropriate videos, (2) After filtering, the action segments of each procedure video were annotated by three members, (3) Finally, cross-checking of annotation results between every two groups was conducted to identify and rectify errors and omissions. This process resulted in a minimum of three annotated action boundaries for each video. To ensure reliable annotations, we employed the complete linkage algorithm to cluster and merge various temporal boundaries into stable boundaries that received multiple agreements. It is important to mention that a single video may feature multiple separate instances of the target action, leading to multiple boundary definitions.

**Is the dataset self-contained, or does it link to or otherwise rely on external resources (e.g., websites, tweets, other datasets)?** If it links to or relies on external resources, a) are there guarantees that they will exist, and remain constant, over time; b) are there official archival versions of the complete dataset (i.e., including the external resources as they existed at the time the dataset was created); c) are there any restrictions (e.g., licenses, fees) associated with any of the external resources that might apply to a future user? Please provide descriptions of all external resources and any restrictions associated with them, as well as links or other access points, as appropriate.

The download source of the dataset can be found on our project homepage, and we provide multiple download options, such as Google Drive and Baidu NetDisk.

**Does the dataset contain data that might be considered confidential (e.g., data that is protected by legal privilege or by doctor-patient confidentiality, data that includes the content of individuals non-public communications)?** If so, please provide a description.

No.

**Does the dataset contain data that, if viewed directly, might be offensive, insulting, threatening, or might otherwise cause anxiety?** If so, please describe why.

No.

**Does the dataset relate to people?** If not, you may skip the remaining questions in this section.

The dataset contains actions performed by nurses.

**Does the dataset identify any subpopulations (e.g., by age, gender)?** If so, please describe how these subpopulations are identified and provide a description of their respective distributions within the dataset.

No.

**Is it possible to identify individuals (i.e., one or more natural persons), either directly or indirectly (i.e., in combination with other data) from the dataset?** If so, please describe how.
No.

**Does the dataset contain data that might be considered sensitive in any way (e.g., data that reveals racial or ethnic origins, sexual orientations, religious beliefs, political opinions or union memberships, or locations; financial or health data; biometric or genetic data; forms of government identification, such as social security numbers; criminal history)?** If so, please provide a description.
Yes, some nursing procedure videos may expose sensitive body parts, such as Change a Two-Piece Pouching System, Female Retention Catheterization, etc.

**Any other comments?** No other comments.

## Collection Process

**How was the data associated with each instance acquired?** Was the data directly observable (e.g., raw text, movie ratings), reported by subjects (e.g., survey responses), or indirectly inferred/derived from other data (e.g., part-of-speech tags, model-based guesses for age or language)? If data was reported by subjects or indirectly inferred/derived from other data, was the data validated/verified? If so, please describe how.
We queried YouTube using text-based searches for each procedure and obtained videos whose titles included the desired procedure keywords. To expand the video collection, we enhanced the search queries by including synonyms of each procedure. For example, *Subcutaneous Injection Insulin* can also be called *Subcutaneous Insulin Administration*, *Subcutaneous Insulin Therapy*, or abbreviated as *SCII*. Each video was downloaded at the highest resolution available. During video retrieval, we prioritized videos shorter than 20 minutes to limit the total storage.

**What mechanisms or procedures were used to collect the data (e.g., hardware apparatus or sensor, manual human curation, software program, software API)?** How were these mechanisms or procedures validated?
We use the Youtube-dl software API to obtain the videos.

**If the dataset is a sample from a larger set, what was the sampling strategy (e.g., deterministic, probabilistic with specific sampling probabilities)?**
No.

**Who was involved in the data collection process (e.g., students, crowdworkers, contractors) and how were they compensated (e.g., how much were crowdworkers paid)?**
In our study, we formed a medical team of 26 individuals, consisting of a nursing lecturer and 25 nursing majors from a medical college, to perform labeling. Over half of the students have at least three years of undergraduate education, possess extensive practical experience in nursing procedures, and have successfully completed the university's standardized nursing procedure assessment.

**Over what timeframe was the data collected? Does this timeframe match the creation timeframe of the data associated with the instances (e.g., recent crawl of old news articles)?** If not, please describe the timeframe in which the data associated with the instances was created.
We collected videos before May 1st, 2023 through keyword search.

**Were any ethical review processes conducted (e.g., by an institutional review board)?** If so, please provide a description of these review processes, including the outcomes, as well as a link or other access point to any supporting documentation.
No.

**Did you collect the data from the individuals in question directly, or obtain it via third parties or other sources (e.g., websites)?**
As described above, the data was collected from Youtube.

**Were the individuals in question notified about the data collection?** If so, please describe (or show with screenshots or other information) how notice was provided, and provide a link or other access point to, or otherwise reproduce, the exact language of the notification itself.
No. The author uploaded video on Youtube for public view, but they did not explicitly know their work would be used in the dataset.

**Did the individuals in question consent to the collection and use of their data?** If so, please describe (or show with screenshots or other information) how consent was requested and provided, and provide a link or other access point to, or otherwise reproduce, the exact language to which the individuals consented.
Followed the CC BY 4.0 license agreement, the author of the videos allows people to remix, transform, and build upon the material for any purpose.

**If consent was obtained, were the consenting individuals provided with a mechanism to revoke their consent in the future or for certain uses?** If so, please provide a description, as well as a link or other access point to the mechanism (if appropriate).
No.

**Has an analysis of the potential impact of the dataset and its use on data subjects (e.g., a data protection impact analysis) been conducted?** If so, please provide a description of this analysis, including the outcomes, as well as a link or other access point to any supporting documentation.
No.

**Any other comments?** No other comments.

## Preprocessing/cleaning/labeling

**Was any preprocessing/cleaning/labeling of the data done (e.g., discretization or bucketing, tokenization, part-of-speech tagging, SIFT feature extraction, removal of instances, processing of missing values)?** If so, please provide a description. If not, you may skip the remainder of the questions in this section.
The annotation work in the early stages is mainly divided into two phases: (1) procedure verification; (2) action localization annotation. In the first phase, annotators need to check the information prompted by the interface, judge whether the video belongs to the nursing procedure, and filter out unqualified videos such as animations, voice broadcasts, and slideshows. In the second phase, the annotation mainly focuses on the action localization of the qualified videos screened in the first phase. The two frame windows correspond to the starting frame and ending frame of the annotated segment.

**Was the "raw" data saved in addition to the preprocessed/cleaned/labeled data (e.g., to support unanticipated future uses)?** If so, please provide a link or other access point to the "raw" data.
We have saved all the data and if needed, it can be obtained by email. However, most of these videos are filtered and considered to be useless.

**Is the software used to preprocess/clean/label the instances available?** If so, please provide a link or other access point.
Yes, we have independently designed annotation software, and annotators can access the data on the cloud through the software for annotation. If needed, please email us for access.

**Any other comments?** No other comments.

## Uses

**Has the dataset been used for any tasks already?** If so, please provide a description.
To evaluate the efficacy of current deep learning methods on nursing activity understanding, we establish three benchmarks on NurViD: procedure recognition on untrimmed videos, procedure and action recognition on trimmed videos, and action detection.

**Is there a repository that links to any or all papers or systems that use the dataset?** If so, please provide a link or other access point.
We plan to create a section on the project homepage to keep track of NurViD-related papers for researchers to analyze and compare.

**What (other) tasks could the dataset be used for?**
We believe that NurVid can also benefit the community of few-shot action understanding classification.

**Is there anything about the composition of the dataset or the way it was collected and preprocessed/cleaned/labeled that might impact future uses?** For example, is there anything that a future user might need to know to avoid uses that could result in unfair treatment of individuals or groups (e.g., stereotyping, quality of service issues) or other undesirable harms (e.g., financial harms, legal risks) If so, please provide a description. Is there anything a future user could do to mitigate these undesirable harms?
Yes.

**Are there tasks for which the dataset should not be used?** If so, please provide a description.
No.

**Any other comments?** No other comments.

## Distribution

**Will the dataset be distributed to third parties outside of the entity (e.g., company, institution, organization) on behalf of which the dataset was created?** If so, please provide a description.
Any institution or organization can conduct academic research for non-profit purposes.

**How will the dataset will be distributed (e.g., tarball on website, API, GitHub)** Does the dataset have a digital object identifier (DOI)?
We will open-source our dataset on our GitHub project homepage. At the moment, we do not have a DOI number.

**When will the dataset be distributed?**
Before June 30, 2023.

**Will the dataset be distributed under a copyright or other intellectual property (IP) license, and/or under applicable terms of use (ToU)?** If so, please describe this license and/or ToU, and provide a link or other access point to, or otherwise reproduce, any relevant licensing terms or ToU, as well as any fees associated with these restrictions.
No.

**Have any third parties imposed IP-based or other restrictions on the data associated with the instances?** If so, please describe these restrictions, and provide a link or other access point to, or otherwise reproduce, any relevant licensing terms, as well as any fees associated with these restrictions.
No.

**Do any export controls or other regulatory restrictions apply to the dataset or to individual instances?** If so, please describe these restrictions, and provide a link or other access point to, or otherwise reproduce, any supporting documentation.
No.

**Any other comments?** No other comments.

## Maintenance

**Who will be supporting/hosting/maintaining the dataset?**
AIM for Health Lab at Monash University.

**How can the owner/curator/manager of the dataset be contacted (e.g., email address)?**
Anyone can contact the manager through the GitHub website message or email.

**Is there an erratum?** If so, please provide a link or other access point.
Not yet.

**Will the dataset be updated (e.g., to correct labeling errors, add new instances, delete instances)?** If so, please describe how often, by whom, and how updates will be communicated to users (e.g., mailing list, GitHub)?
There are no plans at the moment, but if there are updates, they will be announced, and the download source will be updated on the project homepage.

**If the dataset relates to people, are there applicable limits on the retention of the data associated with the instances (e.g., were individuals in question told that their data would be retained for a fixed period of time and then deleted)?** If so, please describe these limits and explain how they will be enforced.
No.

**Will older versions of the dataset continue to be supported/hosted/maintained?** If so, please describe how. If not, please describe how its obsolescence will be communicated to users.
We will not update the dataset in the short term. If there are any updates, the previous version of the dataset will also be shared on Google Drive, Baidu NetDisk, and other platforms for download.

**If others want to extend/augment/build on/contribute to the dataset, is there a mechanism for them to do so?** If so, please provide a description. Will these contributions be validated/verified? If so, please describe how. If not, why not? Is there a process for communicating/distributing these contributions to other users? If so, please provide a description.
We welcome and encourage researchers to extend/augment/build on/contribute to our dataset for non-profit purposes without the need for prior notification.

**Any other comments?** No other comments.