# OpenReview forum: "NurViD: A Large Expert-Level Video Database for Nursing Procedure Activity Understanding"
_NeurIPS.cc/2023/Track/Datasets_and_Benchmarks — NeurIPS 2023 Datasets and Benchmarks Poster_

### Official Review · Reviewer_CHPV · 2023-07-18
**Initial review**

**Rating:** 7
**Confidence:** 4
**Correctness:** Yes.
**Clarity:** Yes.

**Strengths:**

1) The proposed nursing procedure video dataset has advantages over previous related datasets on the data scale, diversity and expert annotations. This can be useful for nursing procedure activity understanding.
2) Comprehensive experiments are conducted on the dataset to build benchmark results for three tasks.

**Additional Feedback:**

None

**Documentation:**

Not sufficient. The reviewer can not access the dataset through the provided URL.

**Limitations:**

Yes.

**Opportunities For Improvement:**

1) There is no data or annotations in the provided dataset Url. Providing real data and annotations (at least a part of them) are more convincing to verify the contributions of the paper.
2) Documentation regarding organization, availability and maintenance of the data is not sufficient.

**Relation To Prior Work:**

Yes.

**Summary And Contributions:**

This paper proposes a video dataset for nursing procedure activity understanding. The nursing procedure videos are collected from Youtube with the guidance of nursing profession. Invalid videos are filtered out and valid videos are annotated with action boundaries by nursing professionals. Benchmark results for three tasks are built based on the proposed dataset.

After author response: I have read the authors' response. Now I am satisfied that the data is available to the reviewers. Considering the overall contribution and impact of the proposed dataset, I decide to increase my rating to accpetance.

---

> ### Author Response · Authors · 2023-08-16
>
> We would like to express our deepest appreciation for the time you dedicated to reviewing our work.
>
> 1.Dataset Release: We apologize for the delayed release caused by the data calibration process. We have shared all the data on the project website (https://github.com/minghu0830/NurViD-benchmark). This dataset includes the original videos, preprocessed videos, RGB, optical flow features, and more. To ensure project reproducibility, we have also shared the corresponding code.
>
> 2.Dataset Governance: In terms of dataset downloads, we have proposed two methods for accessing the data: (1) Following the common practice in most video benchmark tests, we will provide users with video URLs and the code to download the dataset, respecting the rights of YouTube content creators to remove their videos. (2) Users can access the dataset through the Google Drive and Baidu Netdisk download links we have shared, provided they sign the dataset agreement form and comply with the relevant terms of use. We provide complete annotation files, original videos, processed videos, data preprocessing scripts, and dataset splits for different tasks. In terms of code open sourcing and maintenance, we have provided detailed instructions on the usage steps for downloading, preprocessing, feature extraction, and model training codes to ensure the reproducibility of the project. In terms of documentation structure, we have logically organized the directory structure on our project website and provided explanations for the purpose of each document.

---

### Official Review · Reviewer_ysqj · 2023-07-21
**Good, Useful Dataset**

**Rating:** 7
**Confidence:** 5
**Correctness:** Yes, but the dataset itself is yet to…
**Clarity:** Yes, minor typos.

**Strengths:**

1. The dataset is second-largest in size and the most diverse nursing procedure video dataset. This would be helpful to the community.
2. Paper well showcases the dataset and its properties/stats (unlike quite a few other dataset papers which ended up focusing framework development).
3. Annotations are provided by field experts.


**Additional Feedback:**

NA

**Documentation:**

Dataset is not released, yet.

**Limitations:**

Limitations are very briefly, rather superficially, touched upon currently in the paper. This section maybe improved upon.

**Opportunities For Improvement:**

1. Motivation is a bit weak. I am not sure how an action recognition dataset can really help with real-time guidance to nursing students (L28-30)? The need for nursing procedure video dataset is not exactly supported well right now in the introduction.
2. L132: crawlering -> crawling.
3. Dataset is not yet released to the best of my knowledge.

**Relation To Prior Work:**

Yes.

**Summary And Contributions:**

This paper presents second largest and the most diverse dataset for nursing activity/action recognition and detection. Dataset, in all, contains 1538 samples covering 51 types of nursing procedures; 5615 sub-procedure step samples covering 177 nursing action steps. This dataset supports three tasks: procedure recognition; sub-procedure action recognition; and step detection. This paper then reports baseline performance of a couple of methods on the three tasks.

Note (July 20, 2023): Authors mentioned that the dataset will be released by June 30th --- it is already well past that date, and the dataset has still not been released. I am willing to recommend to accepting this paper, given that the authors release the dataset by rebuttal deadline. I am open to rejecting the paper if the dataset is not released.

---

> ### Author Response · Authors · 2023-08-16
>
> Thank you for your insightful review, we appreciate your time and valuable feedback.
>
> 1.Movitation: During the project, we review numerous relevant literature and discuss with nursing professors and students, we've identified two principal challenges within nursing training and learning. (1) Procedure videos retrieval and localization: Students encounter inaccurate recommendations on websites when searching for instructional videos. For example, when searching for Intravenous Injection procedure videos on platforms like YouTube, existing recommendation algorithms might inadvertently suggest Intravenous Blood Sampling procedure videos due to inherent limitations. Our system, trained on the NurViD dataset, yields notably more accurate classification recommendations for nursing-related search queries. Furthermore, NurViD's temporal localization annotations for actions enable dynamic adjustment of playback ranges based on specific interests. This means that specific steps, such as skin disinfection, can be pinpointed within videos, eliminating unnecessary time wastage. (2) Real-time supervision and recording: The substantial number of nursing students in China alone, with over 200,000 new undergraduate students annually, and this figure continues to grow. They must pass a skills exam and complete 1000-2000 hours of practical training. Presently, students heavily depend on experienced instructors for real-time supervision and feedback during training, leading to significant demand for human resources and time allocation. NurViD is a dataset that leans more towards general models and is currently primarily used for testing and improving deep learning models. Our primary intention is to provide assistance for internal learning and training within organizations or institutions. Our focus is on reducing the time and human resources required for independent student learning and facilitating the recording of everyday events through video documentation. Under this mode, rough feedback often saves a significant amount of costs as well.
>
> 2.Typo (crawlering -> crawling): We have corrected this error in the latest manuscript and have also verified the spelling and grammar of other sections.
>
> 3.Dataset Release: We apologize for the delayed release caused by the data calibration process. We have shared all the data on the project website (https://github.com/minghu0830/NurViD-benchmark). This dataset includes the original videos, preprocessed videos, RGB, optical flow features , and more. To ensure project reproducibility, we have also shared the corresponding code. To access the dataset, users have two options. Firstly, they can utilize the provided scripts to download the videos directly from YouTube and then process these videos based on provided scripts and instructions. Secondly, use can directly access our complete data on Google Drive by signing a data agreement form. To facilitate users' understanding and utilization of the dataset, we have made efforts to provide comprehensive documentation and clear instructions. These resources will assist users in navigating the dataset and utilizing its features effectively. We believe that these measures will enhance the overall usability and reproducibility of our project.
>
> 4.Limitations Discussion: We have reexamined the ethical issues and potential biases that the project may bring from multiple perspectives in the Limitation section and Appendix A: Limitations Supplement section. This includes Intended/Foreseeable Uses, Potential Privacy, Employment Risks, Contestability/Explainability Issues, Potential Regional Biases, Comprehensiveness of Nursing Procedures and Actions, among others. Additionally, we have analyzed feasible solutions to mitigate these potential risks and biases.

---

> ### Comment · Reviewer_ysqj · 2023-08-27
>
> I would like to thank authors for addressing my concerns and questions satisfactorily. After considering all the factors, and expecting this dataset to be quite useful, I maintain my original recommendation of accepting this paper.

---

### Official Review · Reviewer_sFtw · 2023-07-21
**Nice dataset for nursing procedure activity understanding**

**Rating:** 6
**Confidence:** 4
**Clarity:** This paper is well organized.

**Strengths:**

There is current lack of sufficiently labelled datasets for the development of automatic recognition systems within nursing procedure activity. Existing datasets face limitations including small-scale size, a focus on single procedures, and a lack of temporally localized annotations.

To address these issues, the authors propose NurViD, a large-scale video dataset with expert-level annotations for understanding nursing procedures. The NurViD dataset, which is about four times longer than the largest existing dataset, consists of over 1.5k videos totaling 144 hours. It covers 51 distinct nursing procedures and 177 action steps, offering a more comprehensive scope compared to previous datasets.

The authors also established three benchmarks on NurViD for evaluating the effectiveness of deep learning methods on nursing activity understanding. These benchmarks involve procedure recognition on untrimmed videos, procedure and action recognition on trimmed videos, and action detection.

To this end, they plan to make the benchmark and code publicly accessible.


**Additional Feedback:**

Nice work!

**Correctness:**

The dataset is constructed in a reasonable way. The benchmark experiments are sufficient.

**Documentation:**

The data collection and organization are clear.

**Ethics:**

No Ethics concern from my perspective

**Limitations:**

For video action classification, one thing that happens for the general benchmark such as kinetics is that the model trained on those datasets sometimes relies more on static images rather than the activity motions to make prediction.  For example, the model can simply classify a person is playing football by simply inputting a single image about the soccer and the playfield. Some dataset such as somethingsomething try to address such bias by including the activities that put more emphasis on the motions. I am wondering what is the case for the proposed dataset.

A brief discussion and some quantitative experiment would be helpful. For example, the author can investigate the influence on the number of frames as input to ablate the model so that it becomes more clear whether the model is making use of more temporal information or just the static image information.

**Opportunities For Improvement:**

Add more comprehensive activity classes and improve the video quality to reduce the bias in the dataset.

**Relation To Prior Work:**

Yes.

**Summary And Contributions:**

Nursing procedure activity understanding is an important task that can enhance the quality and safety of nurse-patient interactions. However, lacking high quality dataset prevent the progress of this field. The author propose a large and well annotated dataset to foster the development of deep learning in enhancing nurse-patient interaction.

---

> ### Author Response · Authors · 2023-08-19
>
> Thank you for dedicating your valuable time to review our work. We truly appreciate your expertise and the thoughtful recommendations you provided.
>
> 1.Expand Classes and Biases: While curating the dataset, several nursing professors have confirmed that the 51 nursing procedures included in our manuscript cover almost all types of operations assessed in nursing practice exams and real clinical environments. The process of selecting nursing procedures and actions is described in Sec. 3.1 titled "Procedure and Action Definition". In the subsequent maintenance and updates, we are more than willing to supplement our database with additional available nursing procedures or actions if we come across them. Also, our dataset potentially inherits some biases caused by regions, races, or backgrounds. We have specified and discussed them  in the Limitation section and Appendix A: Limitations Supplement section.
>
> 2.Motion Information and Ablation Study: In addition to using RGB features, we also incorporated optical flow features, which capture the motion trajectories and dynamic changes of objects in image sequences, providing temporal action information.  Generally speaking, by incorporating optical flow features, models can better understand and utilize the dynamic changes in image sequences, thereby enhancing their ability to model temporal action information. On the other hand, the number of input frames significantly affects video action classification. Fewer frames may lead to information loss and less accurate classification, while more frames provide better action details and improve classification accuracy and differentiation. We conducted an ablation experiment on the I3D model for the procedure classification on trimmed videos task as a reference. The I3D model was trained for 196 epochs with a batch size of 64, using a base learning rate of 0.1. We employed a cosine decay learning rate scheduler with 34 warm-up epochs. We sampled different frames per clip with a sampling rate of 24.
>
> | Frames | Many | Medium | Few | All |
> |-----|-----|-----|-----|-----|
> |&emsp;  4 |&nbsp;  60.5 |&ensp;&ensp;   52.5 | 39.8| 63.1|
> |&emsp;  8 |&nbsp;  63.9 |&ensp;&ensp;   51.5 | 40.5| 64.4|
> |&emsp; 16 |&nbsp;  67.6 |&ensp;&ensp;   49.9 | 32.9| 62.9|
> |&emsp; 32 |&nbsp;  69.1 |&ensp;&ensp;   51.9 | 35.7| 63.3|
>
> From the experimental results, it can be observed that increasing the number of input frames of the model leads to a significant improvement in the classification performance for the "Many" split. However, in the "Medium" and "Few" splits, there is little to no change, and in some cases, a decrease in performance is observed. We analyze that this is due to the presence of common actions across different nursing procedures, such as handwashing, documentation, skin disinfection, and so on. These common action video clips pose certain challenges for both model training and prediction. The model not only needs to consider the temporal information between action frames but also needs to take into account the different environmental contexts, instrument characteristics, and other features associated with different procedures. In scenarios with a large amount of video data, increasing the number of input frames allows the model to learn these features. However, when the number of videos is limited, the model may still struggle to capture these specific features.

---

### Official Review · Reviewer_Y27H · 2023-07-22
**Needed dataset, good benchmarking, but some details are unclear**

**Rating:** 6
**Confidence:** 3

**Strengths:**

1) The dataset is the largest of its kind, and it offers a good variety of actions and procedures that are relevant and important for nursing procedure.
2) The level of annotation includes high-level and fine-grained annotations which makes the dataset useful for various tasks
3) The annotation was done by experts in the domain and students in the field of nursing who have passed a standardized test in their studies.
4) The type of videos used makes it more realistic as they comes from actual nursing procedures from youtube.

**Additional Feedback:**

The paper is well written, and it presents an amazing efforts to enable the applications of computer vision in nursing. The authors based their work on guideline of Training Outline for Newly Employed Nurses issued by the National Health Commission of China. However, it could be worth mentioned the relevance and discrepancies between different nursing procedures around the globe (if any).

**Clarity:**

Overall, the paper is well written. However, in the split of the frequency of actions and procedures, the authors opted for a three-class classification including "many, medium, and few" to represent the occurrence of an action.  This split changes from a task to another which was not clear to me why.

In addition, in line 201, the authors included this statement: "To account for the long-tailed nature of the NurViD dataset, we categorized the procedure classes into three groups based on their frequency: many, medium, and few. The classification into these categories was determined by calculating the average per-class accuracy for each category, taking into account the frequency percentiles."

This was not clear to me why would accuracy be used to determine frequency while it could be determined statistically from the dataset.  In addition, I find such classification is general is not needed since part of the challenge in this dataset to classify classes that are not well-represented in the collected dataset.



**Correctness:**

The claims in this work appear correct to me and are supported by evidence. The benchmark results are fair and are reported in a nice way. However, the split of data is unclear and could need correction. This will be discussed in the next section.

**Documentation:**

The details of the dataset are somewhat sufficient. However, the train-validation-test splits was not clear to me if it is predetermined or random. If predetermined, the selection procedure must be detailed. If random, this could cause ambiguities in reporting results for future users of the dataset.

**Ethics:**

Given that the videos are collected from youtube, the license must be carefully studied to ensure that using and reproducing such videos is permissible. This is especially needed since all videos are collected from youtube with human subjects appearing in them.

**Limitations:**

The authors have mentioned and discussed the limitations of the presented dataset in the limitations section in an adequate manner. However, the authors are advised to include a discussion of limitations from a nursing procedure point of view such as missing procedures/actions (if any).

**Opportunities For Improvement:**

1- The split of videos (many, medium, and few) could be done in a better standardized way to ensure the same criterion is applied accord the board for the three tasks included in the dataset.

2- Including videos with no procedures/action to ensure that models trained on such dataset do not always assume that every input must have a nursing procedure.  It is advised that videos from the same "hospital environment" are included for normal activities that are not nursing procedure. Note that the authors have mentioned that videos that are "4) lacking the specified procedure" have been filtered out which I believe should not have been done.

**Relation To Prior Work:**

The authors have included a good literature review that I find sufficient. However, for procedure classification task, the use of activity recognition models might not be appropriate. I would advise the authors to include a section on the models and the motivation behind using activity recognition models for benchmarking. In addition, a discussion of what type or class of models could be used or needed to perform each task.

**Summary And Contributions:**

The authors of this work present a nursing video dataset with different nursing procedures. The presented dataset is claimed to be the largest dataset of its kind. The authors present the dataset with three different tasks that are: 1) procedure classification on untrimmed videos, 2) Procedure and action classification on Trimmed videos, and 3) action detection on untrimmed videos.
The dataset is obtained from YouTube, and manually annotated by at least 3 people with experience/education in the field of nursing.

The main contributions of this work are:
1) A large open-source annotated video dataset for nursing procedures
2) Identifying 3 different challenging tasks on the dataset
3) A benchmark analysis of the dataset on the defined tasks

---

> ### Author Response · Authors · 2023-08-16
>
> We would like to extend our heartfelt thanks to you for the time you spent reviewing our research. Your constructive feedback has been incredibly helpful in refining our study.
>
> 1.The Split of Videos: We apologize for any confusion caused by ambiguous statements. (1) Split method: Our task is to classify 51 procedures and 171 actions. Procedure classification is based on untrimmed videos, while action videos are based on trimmed video segments. In order to solely observe the impact of the long-tail distribution of data on model performance, we categorized them as many, medium, and low based on the quantity of procedure untrimmed videos and action untrimmed videos. The accuracy results still represent the average accuracy of procedures and actions within each split. The split is fixed to allow researchers to replicate and compare model results. (2) Unified split: A nursing operation video often comprise multiple action segments with uneven distribution, it is impractical to use a unified split for these two tasks. We have clearly defined the split for each category in the file located at /annotations/Procedure&Action_ID.xlsx file on the dataset release website. Additionally, we have modified the statements in the manuscript to address the ambiguitiy you mentioned.
>
> 2.Video Filtering: In the process of curating the dataset, we follow the most widely acknowledged action recognition video datasets, such as ActivityNet. These benchmarks exclude any unrelated elements during curating. However, we agree that incorporating irrelevant action segments can potentially improve the dataset's quality and make it a more realistic setting. To further enhance research opportunities, we are dedicated to open-sourcing the remaining validated videos and categorizing them as "Others" for further research purposes.
>
> 3.Copyright and License: According to YouTube's copyright exception policy[1], CC-BY-4.0[2] and the copyright disclaimer[3] under Sec. 107 of the Copyright Act 1976 (the "fair use doctrine"): copyrighted material may be used for non-commercial purposes such as scholarship, and research. Furthermore, we have proposed two methods for access the data: (1)we following the practice commonly used in most video benchmark tests (ActivityNet[4], YouTube-8M[5], Kinetics 400/600/700[6], etc.), we provide users with video URLs and code to download the dataset, respecting the rights of YouTube content creators to remove their videos. (2)users can utilize the Google Drive and Baidu Netdisk download links we have shared, provided that they sign the dataset agreement form and comply with the relevant terms of use.
>
> 4.Potential Regional Biases: Given the diverse global origins of the videos and the inherent variance in nursing procedure guidelines across countries, standardization emerges as a formidable challenge. To address this issue, while referring to the action step standards in the "Training Outline for Newly Employed Nurses" guidelines issued by China's National Health Commission, we also watched the collected videos to summarize more universally applicable action labels. We had continuous involvement from nursing professors who provided feedback. According to their feedback, NurViD covers almost all common action labels, and different regions can choose their own standards based on it. However, despite our efforts, it is important to acknowledge that bias cannot be completely avoided. Therefore, we have provided a detailed discussion in the Limitation section and Appendix A: Limitation Supplement section.
>
> 5.Limitations of the Comprehensiveness of Nursing Procedures and Actions:
> We initially selected 60 common nursing procedures and 211 actions based on references from the Nurselabs website and four nursing procedure books. However, after discussions with nursing experts, we concluded that some nursing procedures, such as enemas and abdominal lavage, are relatively less common. Moreover, the current set of 51 nursing procedures already encompasses nearly all commonly used operations in clinical environments. We also took into consideration the quantity and quality of the retrieved videos. As a result, we decided to remove 9 nursing procedures and 40 actions that were deemed unsuitable. If there are suitable nursing procedures for future maintenance, we will continue to update them. We have discussed this limitation in the Limitation section and Appendix A: Limitation Supplement section.

---

> ### Author Response · Authors · 2023-08-21
>
> Just to add: In Appendix D, "The Long-tail Distribution of Procedure, Action, and Their Composition," we also describe the criteria for our split and provide line graphs for better visual understanding.

---

### Author Response · Authors · 2023-08-16

We sincerely appreciate the reviewers' thoughtful and constructive evaluation of our manuscript, acknowledging their dedicated time and effort in meticulously assessing our work. After carefully considering the feedback, we have taken note of several general concerns raised by the reviewers. Therefore, we respond to them here in a unified manner. We will also respond to questions raised by each reviewer individually. If you have any further questions or comments regarding ethics, technology, or any related matters, please let us know. We are happy to follow up!

1.Dataset Release: We sincerely apologize for the delay in releasing the data, which was caused by the data calibration process. We have shared all the data on the project website (https://github.com/minghu0830/NurViD-benchmark). This dataset includes the original videos, preprocessed videos, RGB, optical flow features, and more. To ensure project reproducibility, we have also shared the corresponding code and annotation files. To access the dataset, users have two options. Firstly, they can utilize the provided scripts to download the videos directly from YouTube and then process these videos based on provided scripts and instructions. Secondly, users can directly access our complete data on Google Drive by signing a data agreement form. To facilitate users' understanding and utilization of the dataset, we have made efforts to provide comprehensive documentation and clear instructions. These resources will assist users in navigating the dataset and utilizing its features effectively. We believe that these measures will enhance the overall usability and reproducibility of our project.

2.Copyright and License: According to YouTube's copyright exception policy[1], CC-BY-4.0[2] and the copyright disclaimer[3] under Sec. 107 of the Copyright Act 1976 (the "fair use doctrine"): copyrighted material may be used for non-commercial purposes such as scholarship, and research. Furthermore, we have proposed two methods for access the data: (1)we following the practice commonly used in most video benchmark tests (ActivityNet[4], YouTube-8M[5], Kinetics 400/600/700[6], etc.), we provide users with video URLs and code to download the dataset, respecting the rights of YouTube content creators to remove their videos. (2)users can utilize the Google Drive and Baidu Netdisk download links we have shared, provided that they sign the dataset agreement form and comply with the relevant terms of use. For the code, the project follows the Apache License 2.0[7].

3.Ethical Issues and Potential Biases: We have reexamined the ethical issues and potential biases that the project may bring from multiple perspectives in the Limitation section and Appendix A: Limitations Supplement section. This includes Intended/Foreseeable Uses, Potential Privacy, Employment Risks, Contestability/Explainability Issues, Potential Regional Biases, Comprehensiveness of Nursing Procedures and Actions, among others. Additionally, we have analyzed feasible solutions to mitigate these potential risks and biases.

Ref:

[1] YouTube's copyright exception policy: https://www.youtube.com/howyoutubeworks/policies/copyright/\#copyright-exceptions

[2] CC-BY-4.0: https://creativecommons.org/licenses/by/4.0/

[3] Copyright disclaimer: https://support.google.com/youtube/thread/113884705/copyright-disclaimer-under-section-107-of-the-copyright-act-1976-allowance-is-mad-for-fair-use-f

[4] S. Abu-El-Haija, N. Kothari, J. Lee, P. Natsev, G. Toderici, B. Varadarajan, and S. Vijayanarasimhan. Youtube-8m: A large-scale video classification benchmark, 2016.

[5] Fabian Caba Heilbron, Victor Escorcia, Bernard Ghanem, and Juan Carlos Niebles. Activitynet: A large-scale video benchmark for human activity understanding. In Proceedings of the ieee conference on computer vision and pattern recognition, pages 961–970, 2015.

[6] L. Smaira, J. Carreira, E. Noland, E. Clancy, A. Wu, and A. Zisserman. A short note on the kinetics-700-2020 human action dataset, 2020.

[7] Apache-2.0 License: http://www.apache.org/licenses/LICENSE-2.0.html

---

### Decision · Program_Chairs · 2023-09-22

**Decision:**

Accept (Poster)

**Comment:**

This paper receives four unanimous recommendations: four accepts. The reviewers all think this paper is valuable for video understanding. After a careful check, the AC agrees with the reviewers, and thus makes an accept recommendation to this paper.